# Training-Free Multimodal Large Language Model Orchestration

## Abstract

Different Multimodal Large Language Models (MLLMs) cannot be integrated into a unified multimodal input-output system directly. In previous work, training has been considered as an inevitable component due to challenges in modal alignment, Text-to-Speech efficiency and other integration issues. In this paper, we introduce Multimodal Large Language Model Orchestration (MLLM Orchestration), an effective approach for creating interactive multimodal AI systems without additional training. MLLM Orchestration leverages the inherent reasoning capabilities of large language models to coordinate specialized models through explicit workflows, enabling natural multimodal interactions while maintaining modularity, improving interpretability, and significantly enhancing computational efficiency. Our orchestration framework is built upon three key innovations: (1) a central controller LLM that analyzes user inputs and dynamically routes tasks to appropriate specialized models through carefully designed agents; (2) a parallel Text-to-Speech architecture that enables true full-duplex interaction with seamless interruption handling and natural conversational flow; and (3) a cross-modal memory integration system that maintains coherent context across modalities through intelligent information synthesis and retrieval, selectively avoiding unnecessary modality calls in certain scenarios to improve response speed. Extensive evaluations demonstrate that MLLM Orchestration achieves comprehensive multimodal capabilities without additional training, performance improvements of up to 7.8% over traditional jointly-trained approaches on standard benchmarks, reduced latency by 10.3%, and significantly enhanced interpretability through explicit orchestration processes. Our work establishes orchestration as a practical alternative to joint training for multimodal systems, offering greater efficiency, adaptability, and transparency for next-generation AI interactions.

## 1 Introduction

Recent advances in Large Language Models (LLMs) OpenAI et al. (2023); Team et al. (2023); Grattafiori et al. (2024); Liu et al. (2024a) have enabled sophisticated multimodal capabilities. GPT-4o Hurst et al. (2024) demonstrated the feasibility of processing multiple modalities simultaneously, sparking interest in omni-modal models. This pursuit aligns with human intuition - seamlessly integrating visual, auditory, and textual information. Such unified processing offers more fluid interactions and comprehensive understanding by leveraging complementary information across modalities Zhang et al. (2023); Chen et al. (2024b); Wang et al. (2024b); Tong et al. (2024); Fu et al. (2024).

The development of omni-modal capabilities has progressed through several key technical advances. Following GPT-4o's success, research efforts have primarily focused on two core aspects: modality expansion and natural interaction enhancement. For modality expansion, early attempts like MiniGPT-4 Zhu et al. (2023) established foundational techniques through a two-stage alignment approach, using a pre-trained BLIP-2 visual encoder and a lightweight projection layer. LLaVA-NeXT Liu et al. (2024a) further extended this by introducing a unified visual representation learning framework, while LLaMA-Omni Fang et al. (2024) and RLAIF-V Yu et al. (2024) proposed novel architectures for handling diverse modalities. For natural interaction enhancement, as demonstrated in Figure 1 (a): VITA Fu et al. (2024) pioneered non-awakening interaction and audio interrupt handling capabilities through a three-stage training pipeline (bilingual instruction fine-tuning, mul-

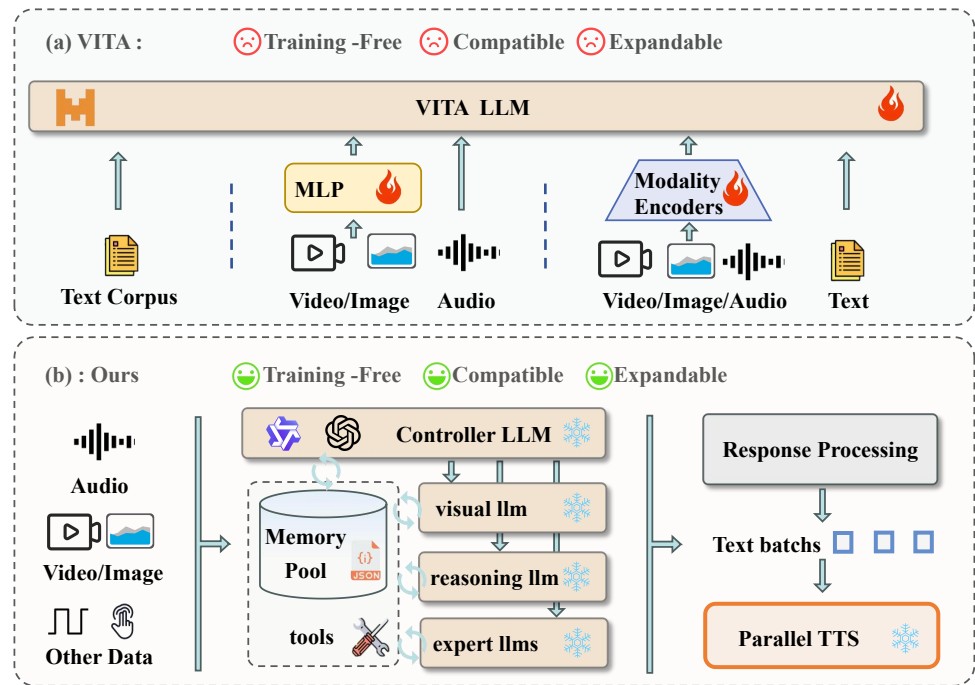

Figure 1: illustrates the training procedures of VITA(a) and our Training-Free Multimodal Large Language Model Orchestration (b). Our framework presents a training-free and efficient pipeline for handling omni data.

timodal alignment training, and multimodal instruction fine-tuning) , while HumanOmni Zhao et al. (2024) focused on human-centric scenarios .

However, existing omni-modal methods Zhu et al. (2023); Liu et al. (2023); Team et al. (2023); Liu et al. (2024a); Shin et al. (2024); Alayrac et al. (2022); Zhao et al. (2024); Fang et al. (2024); Grattafiori et al. (2024) predominantly rely on retraining and expanding a single base model to accommodate multiple modalities. This paradigm suffers from two critical limitations. First, it incurs substantial training costs: aligning heterogeneous modalities necessitates extensive customized datasets and intensive fine-tuning, resulting in significant human effort and computational overhead. Second, it exhibits poor extensibility: modifying the base model or incorporating new modalities generally requires complete retraining, severely limiting rapid adaptation to new scenarios and evolving user demands.

To overcome these limitations, we explore an *agent-based approach* for omni-modal capabilities without retraining. By integrating specialized models dynamically, we enable seamless multimodal interactions while addressing three key challenges: (1) **Assignment**: Dynamically routing tasks to suitable specialized models; (2) **Memory**: Sharing context among heterogeneous models; and (3) **Efficiency**: Ensuring responsive interaction despite coordination overhead. Our training-free MLLM Orchestration framework (Figure 2) integrates three components: a central **Controller LLM** that analyzes user intent and dispatches tasks to expert models; a unified **Cross-modal Memory** storing structured interaction histories in standardized JSON format; and a **Parallel Text-to-Speech (TTS)** architecture reducing latency through semantic-based segmentation. These components enable interpretable, extensible, and responsive multimodal interactions, offering users a unified "super-model" experience composed of specialized expert subsystems while avoiding the overhead and rigidity of training-based approaches.

The main contributions can be summarized as follows:

- **Training-free Orchestration Framework.** A novel multimodal orchestration paradigm that enables efficient interaction through intelligent scheduling and coordination, eliminating the need for extensive training or large datasets.

- **Cross-modal Memory Integration.** A memory integration mechanism that unifies multimodal information into textual representations, enabling seamless context sharing across modalities without training requirements.

- **Parallel Batch TTS Processing.** A high-performance TTS architecture achieving near real-time response through intelligent chunking and buffering, significantly reducing perceived latency while maintaining output quality.

- **Experimental Results.** Our framework achieves comparable or superior performance to state-of-the-art training-based methods (e.g., +1.93% on MMbench, +2.73% on Ai2d), while reducing latency by 10.3% through parallel processing optimizations.

## 2 RELATED WORK

### 2.1 OMNI-MODAL TRAINING AND MULTIMODAL ALIGNMENT

In recent years, Multimodal Large Language Models (MLLMs) have made significant progress with the support of end-to-end training techniques, leading to the emergence of two main technical approaches in training paradigms: full-parameter training and parameter-efficient training. Among these, VITAFu et al. (2024), as the first open-source interactive omni-modal large language model, adopted an innovative three-stage training process that includes bilingual instruction fine-tuning, multimodal alignment training, and multimodal instruction fine-tuning. In terms of parameter-efficient training, Freeze-OmniWang et al. (2024d) proposed a novel freezing training strategy that maintains fixed language model parameters while only training modal adapters. This method not only significantly reduces computational resource requirements but also effectively avoids catastrophic forgetting. Similarly, works like Mini-Omni2Xie & Wu (2024), LLaMA-OmniFang et al. (2024), and MoshiDéfossez et al. (2024) have adopted similar parameter-efficient training strategies, providing important references for reducing the costs associated with multimodal training.

### 2.2 AGENT-BASED MULTIMODAL SYSTEMS

In the field of agent systems, a series of innovative research works have recently emergedWu et al. (2023); Kumar et al. (2024); Liu et al. (2024b); Chen et al. (2023); Han et al. (2024). These works mainly focus on two directions: enhancing multimodal capabilities and optimizing interaction experience. AutoGenWu et al. (2023) proposed a multi-agent dialogue framework that achieves automatic decomposition and collaborative processing of complex tasks through flexible agent interaction patterns and behavioral strategies. mmctagent Kumar et al. (2024) designed a novel multimodal agent architecture that enhanced agent decision-making capabilities in complex visual scenarios through deep alignment of vision-language-behavior and cross-modal reasoning mechanisms. LLaVA-PlusLiu et al. (2024b) explored the tool learning paradigm for agents, proposing a progressive tool discovery and usage mechanism that enables agents to autonomously select and combine appropriate tools based on task requirements.

In agent orchestration, CrewAIDuan & Wang (2024) focuses on role-based agent orchestration, supporting collaboration and task allocation among multiple agents. TaskWeaverQiao et al. (2023) provides an agent-based task automation framework, enabling more flexible workflow management. In professional domain applications, LawLuoSun et al. (2024) developed a multi-round dialogue collaborative framework , simulating real legal consultation scenarios through four professional agents; In intelligent orchestration and optimization, Self-Organized AgentsIshibashi & Nishimura (2024) explored an LLM multi-agent framework for ultra-large-scale code generation and optimization, CMATLiang et al. (2024) successfully enhanced small language model performance through multi-agent collaboration.

In contrast, our proposed system adopts a different approach from traditional agent orchestration, focusing on LLM's intelligent orchestration mechanism. Through an innovative multimodal LLM orchestration framework, the system has achieved integration of video, image, text, and audio modalities, demonstrating high efficiency, flexibility, and scalability. Compared to existing agent systems, our solution is more lightweight and efficient, requiring no complex agent collaboration mechanisms or training, directly achieving multimodal capability coordination through LLM orchestration. Through comprehensive open-sourcing of the system, we hope to provide new research direc-

tions for the multimodal LLM orchestration field and promote the application of this technology in broader practical scenarios.

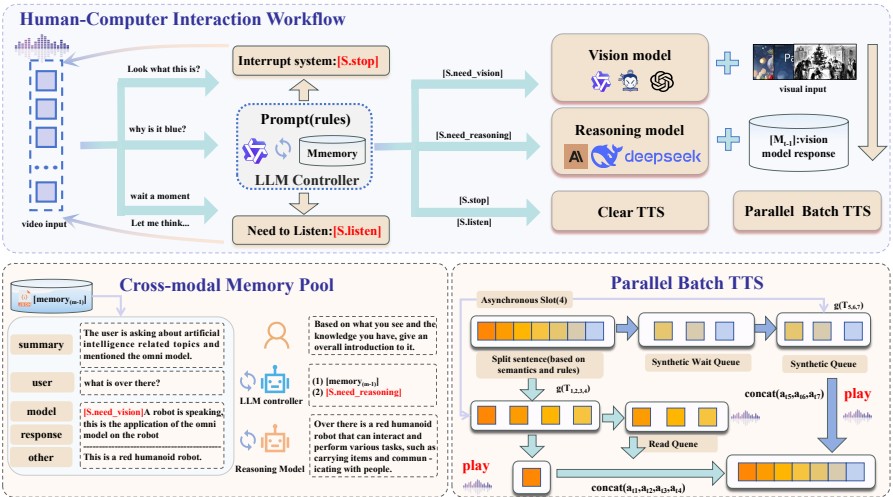

Figure 2: Overview of the MLLM Orchestration framework, featuring core components such as the Central Controller LLM (text generation, vision LLM, and specialized LLMs), multimodal memory integration system, and parallel text-to-speech synthesis mechanism. The training-free pipeline of MLLM Orchestration is transparent, providing good interpretability.

## 3 METHOD

Our MLLM Orchestration framework enables seamless multimodal interaction through a training-free approach. The method consists of three primary modules: **MLLM Orchestration Reasoning**, **Cross-modal Memory Integration**, and **Parallel Text-to-Speech (TTS) Generation**. These modules work collaboratively to achieve efficient task orchestration and multimodal fusion.

**Problem Definition.** Given a user query $q_t$ at turn $t$, the system must generate both a natural language response $o_t$ and an audio output $a_t$. Let $M_{t-1}$ denote the accumulated memory storing relevant multimodal information. The system operates as:

$$F : (q_t, M_{t-1}) \mapsto (o_t, a_t) \tag{1}$$

where $a_t = \text{TTS}(o_t)$ represents speech synthesis of the text response.

As shown in Equation (1), our system takes user query $q_t$ and historical memory $M_{t-1}$ as input, and generates both text response $o_t$ and audio response $a_t$. Unlike previous training-based approaches, we propose a training-free orchestration approach that leverages the inherent reasoning and planning capabilities of a controller LLM. Our method processes inputs through three key orchestration steps: (1) the controller LLM analyzes user intent and selects appropriate specialized models through control token generation, (2) the cross-modal memory integration module retrieves and integrates relevant historical information from $M_{t-1}$ using semantic similarity matching, and (3) the parallelized TTS module efficiently generates the final audio response $a_t$ through semantic segmentation and batch processing. This modular design enables dynamic orchestration of specialized models without requiring end-to-end training.

### 3.1 MLLM ORCHESTRATION

After receiving the user input $q_t$, the first step in our orchestration workflow involves the **controller LLM**. This controller is responsible for interpreting the user's current intent based on both the immediate query and the accumulated cross-modal context $M_{t-1}$, forming the input $x_t = (q_t, M_{t-1})$. It outputs an ordered token sequence:

$$Y_t = f_{\text{ctrl}}(x_t) = [y_1, y_2, \dots, y_L] \tag{2}$$

where each token $y_i$ can either be a content token (for natural language response) or a special control token that drives orchestration behavior. We define the content function $\mathcal{C} : Y_t \to C_t$ that extracts content tokens and the control function $\mathcal{S} : Y_t \to S_t$ that extracts control tokens. The controller's output sequence $Y_t$ can be decomposed as:

$$Y_t = C_t \sqcup S_t \tag{3}$$

where $C_t = \mathcal{C}(Y_t)$ denotes the ordered content tokens and $S_t = \mathcal{S}(Y_t) \subseteq \mathcal{S}_{\text{vocab}}$ represents the set of control tokens, with $\mathcal{S}_{\text{vocab}}$ being the predefined vocabulary of special control tokens (e.g., [S.need_vision], [S.stop], [S.listen]). The operator $\sqcup$ denotes ordered concatenation preserving token positions.

**Control Token Design.** To enable training-free orchestration capability, we design a structured control token vocabulary. Our control tokens follow the format [S.action_modality], where core tokens include: [S.need_vision] for visual analysis, [S.need_reasoning] for logical tasks, [S.listen] for awaiting user input, and [S.stop] for interruption handling. The controller is guided by a dynamic prompt template that incorporates available expert models and their capabilities.

**Prompt Construction.** Our system employs a prompt composer that dynamically constructs controller instructions based on: (1) current dialogue context, (2) available expert model registry, and (3) user query characteristics. The base template includes sections for available tools, expected output format, and behavioral guidelines for different interaction states (listening, speaking, waiting). For instance, when the controller recognizes that the user's question refers to a visual object, it may output:

$$S_t = \{[\text{S.need\_vision}]\} \tag{4}$$

which activates the visual model module in the next phase. The complete algorithm of our MLLM Orchestration framework is detailed in Algorithm 1 in Appendix A.

## 3.2 CROSS-MODAL MEMORY INTEGRATION

Once the controller LLM has determined the required modalities via the output control token set $S_t \subseteq \mathcal{S}$, the orchestration process proceeds by activating the corresponding expert models. These models require access to relevant multimodal inputs, either provided directly by the user or stored from previous turns. To support this, we design a **cross-modal memory pool** that serves as a unified knowledge base for storing and retrieving structured multimodal context across the entire dialogue session.

At each turn $t$, when the controller outputs special tokens such as [S.need_vision], [S.need_speech], or other modality-specific instructions, the system must determine which expert models to invoke. We formalize this process using a modality selection function:

$$\delta_{\text{modality}}(x_t) = \begin{cases} \{m_1, \ldots, m_k\}, & [S.need\_m_i] \in f_{\text{ctrl}}(x_t), \\ \emptyset, & \text{otherwise,} \end{cases} \tag{5}$$

where $m_i \in \mathcal{M}$ is a registered modality (e.g., vision, reasoning, audio), and $\mathcal{M}$ denotes the set of available expert modules in the system. This function maps the controller's output to a set of concrete modality requirements for the current interaction.

For each selected modality $m_i \in \delta_{\text{modality}}(x_t)$, a retrieval function $h_{m_i}$ is invoked to extract the most relevant context or input data from the cross-modal memory pool $M_{t-1}$. For example, when visual information is needed, the system computes:

$$v_t = h_{\text{vision}}(q_t, M_{t-1}) \tag{6}$$

where $v_t$ represents the retrieved visual data (e.g., image embeddings, scene descriptions, or OCR results) relevant to the current query. For generality, we denote the set of all retrieved data as $\{d_t^1, d_t^2, \ldots, d_t^k\}$, where each $d_t^i$ corresponds to a modality-specific data unit.

The retrieved data is integrated into the LLM's reasoning flow using a modality-aware integration function:

$$\widetilde{Y}_t = I(Y_t, \{d_t^1, d_t^2, \ldots, d_t^k\}) \tag{7}$$

where $\widetilde{Y}_t$ is the updated token sequence that replaces placeholder control tokens with actual data descriptions or summaries. Accordingly, the final generated answer becomes:

$$O_t = C_t(\{d_t^1, d_t^2, \ldots, d_t^k\}) \tag{8}$$

which denotes the complete response content informed by multimodal inputs.

**Memory Pool Structure.** The cross-modal memory pool is updated at every turn $t$ to include new multimodal observations, system actions, and dialogue entries. Each memory item follows a structured format:

$$m_i = \{\text{timestamp}, \text{modality}, \text{content}, \text{relevance\_score}\} \tag{9}$$

where entries are indexed by semantic similarity and temporal proximity. This enables efficient storage and retrieval of multimodal information while maintaining extensibility for future modalities.

**Memory Compression.** As the conversation progresses, the memory pool $M_t$ expands, which can eventually exceed the LLM's input length constraints. To address this, we implement a content-aware compression strategy with three key principles: (1) *Recency*: more recent interactions receive higher preservation priority, (2) *Relevance*: semantically similar content to current queries is retained, and (3) *Diversity*: maintaining representation across different modalities. We define a memory compression function:

$$M_t' = h_{\text{compress}}(M_t, \lambda_{\text{rec}}, \lambda_{\text{rel}}, \lambda_{\text{div}}) \tag{10}$$

which outputs a condensed memory $M_t'$ that retains essential information for future reasoning. The compression is guided by weighted factors for recency ($\lambda_{\text{rec}}$), relevance ($\lambda_{\text{rel}}$), and diversity ($\lambda_{\text{div}}$). For instance, older image descriptions and their related QA pairs may be summarized into abstracted forms, while trivial or redundant entries are discarded. In summary, the cross-modal memory module serves as a critical interface for knowledge persistence, dynamic context fusion, and efficient memory management, enabling the system to perform long-range multimodal reasoning without retraining or data loss.

## 3.3 PARALLEL BATCH TTS GENERATION

Parallel batch TTS is a critical component for system feedback to human users. Therefore, we designed a segmentation strategy based on semantic analysis, which divides a complete sentence into different batches according to both semantic coherence and predefined rules. These batches are processed in parallel, significantly reducing the overall synthesis time. This approach ensures that there are no semantic interruptions during the output of complete meanings, thereby enhancing the fluency and naturalness of the audio output.

To optimize for latency and fluency, the system adopts a segmentation-based batch synthesis approach. Upon receiving the finalized content tokens $C_t$, we first apply a rule-based segmentation strategy that divides text at natural prosodic boundaries (commas, periods, clauses) to produce semantically coherent chunks:

$$T = \text{segment}(C_t) = [T_1, T_2, \ldots, T_n] \tag{11}$$

where each segment $T_i$ represents a self-contained phrase or clause. Our segmentation rules prioritize: (1) semantic completeness (avoiding mid-phrase breaks), (2) optimal length (10-50 characters per segment), and (3) prosodic naturalness (respecting punctuation boundaries).

The TTS function $g(T_i)$ is applied to each segment $T_i$ independently and in parallel:

$$a_{t,i} = g(T_i) \quad \text{for } i = 1, 2, \ldots, n \tag{12}$$

To minimize latency, we employ a streaming synthesis and playback strategy. As soon as the first segment $a_{t,1}$ is synthesized, it begins playback immediately while subsequent segments are being synthesized asynchronously:

$$\text{play}(a_{t,i}) \| \{g(T_{i+1}), g(T_{i+2}), \ldots, g(T_n)\} \tag{13}$$

where $\|$ denotes parallel execution. The final audio stream $a_t$ is constructed through real-time concatenation with prosodic adjustments at segment boundaries:

$$a_t = \text{stream\_concat}(a_{t,1}, a_{t,2}, \ldots, a_{t,n}) \tag{14}$$

This streaming approach significantly reduces the perceived latency as users begin hearing the response while the system continues to synthesize remaining segments. The system maintains a buffer of synthesized segments to ensure smooth playback transitions, while the stream_concat operation handles real-time prosodic adjustments to maintain natural speech flow across segment boundaries.

### 3.4 EXAMPLE WORKFLOW

Consider a scenario where a user shows an image of their garden and asks "What flowers are blooming in this image?" At turn $t_1$, the controller LLM analyzes the query and outputs $f_{\text{ctrl}}(q_{t_1}, M_{t_1-1}) \rightarrow \{[\texttt{S.need\_vision}]\}$, triggering the visual model. The system retrieves visual information $v_{t_1} = h_{\text{vision}}(q_{t_1}, M_{t_1-1})$ and identifies various flowers. The response is generated and segmented as $T_{t_1} = \text{segment}(C_{t_1})$, producing "I can see several roses and tulips in full bloom" which enters the TTS pipeline.

While the system is speaking, the user interrupts with "How many roses..." The controller immediately detects the interruption pattern and executes $f_{\text{ctrl}}(q_{t_2}, M_{t_2-1}) \rightarrow \{[\texttt{S.stop}]\}$, followed by $\text{clear}(Q_{\text{TTS}})$ to stop the current speech output. As the question is incomplete, it also outputs $[\texttt{S.listen}]$ and updates the memory $M_{t_2} = M_{t_2-1} \cup \{(q_{t_2}, v_{t_1})\}$.

The user completes their question "...how many roses are there?" The system processes this follow-up query using the cached visual information from memory $v_{t_1} \in M_{t_2}$, without needing to reactivate the vision model. The response "There are 3 red roses in the image" is synthesized through parallel TTS, where $\text{play}(g(T_{t_3,1})) \| g(T_{t_3,2})$ enables immediate playback while preparing subsequent segments.

This natural interaction flow demonstrates how the system seamlessly integrates visual processing ($h_{\text{vision}}$), memory management ($M_t$), and streaming speech synthesis while maintaining responsive user interaction through interrupt handling ($\text{clear}(Q_{\text{TTS}})$).

## 4 EXPERIMENTS AND RESULTS

### 4.1 EXPERIMENTAL SETUP

Our experiments validate training-free orchestration superiority, framework flexibility, and pipeline explainability. We use Qwen2.5-14BChu et al. (2024) as controller and various executors: Qwen2.5-VLBai et al. (2025); Yang et al. (2024a), LLaVA-VideoZhang et al. (2024b), and othersZhang et al. (2024a); Guo et al. (2025); Li et al. (2024).

**Baselines.** We compare against: (1) Commercial solutions: GPT-4oHurst et al. (2024), Claude 3.5Anthropic (2024), Gemini-1.5-ProTeam et al. (2024); (2) Open-source omni-models: Qwen2.5-Omni, VITA, M2-omni; (3) Specialized multimodal models used in isolation.

**Evaluation.** We evaluate on general understanding (MMEFu et al. (2023)), vision tasks (MM-StarChen et al. (2024a), LVBenchWang et al. (2024c)), temporal reasoning (MMMUYue et al. (2024)), and specialized domains (MathVisionWang et al. (2024a), CC-OCRYang et al. (2024b)).

### 4.2 COMPARISON WITH MAINSTREAM OMNI MODELS

Table 1 compares our framework with state-of-the-art omni-modal models. Our approach achieves competitive performance: on MMStar, we reach **69.37%**, exceeding GPT-4o (64.70%) by 4.67% and Qwen2.5-Omni (64.00%) by 5.37%. On MMMU, we achieve **70.04%**, improving over GPT-4o and Qwen2.5-Omni (both 59.20%) by 10.84%. While Gemini-1.5-Pro shows superior Video-MME performance (75.00% vs 65.58%), our method demonstrates competitive results with training-free modularity.

> **Finding 1:** Training-free orchestration achieves competitive performance compared to commercial and open-source omni-modal models while providing superior modularity and interpretability.

| Model | General | Vision | Temporal | Efficiency |
|-------|---------|--------|----------|------------|
|       | Video-MME | MMStar | MMMU | Time (s) |
| GPT-4o | 71.90 | 64.70 | 59.20 | 1.2 |
| Qwen2.5 | 64.30 | 64.00 | 59.20 | 6.0 |
| VITA | 59.20 | 46.40 | 47.30 | 3.7 |
| IXC2.5 | 60.60 | – | – | – |
| M2-omni | 60.40 | 60.50 | 51.20 | – |
| **Ours** | **65.58** | **69.37** | **70.04** | 3.2 |

Table 1: Comparison with leading multimodal models across general understanding, vision, temporal reasoning, and efficiency metrics.

## 4.3 COMPARISON WITH MULTIMODAL LLM

We further compare our method with various other multimodal models to validate its effectiveness across diverse scenarios.

| Model | General Multimodal | | | Vision Understanding | | |
|-------|------|-----------|-----------|--------|---------|-----------|
|       | MME | MMBench-EN | MMBench-CN | MMStar | LVBench | Video-MME |
| Qwen2.5-VL-7B | 1673 | 84.45 | 84.98 | 59.94 | 45.30 | 56.62 |
| Qwen2.5-VL-32B | 1915 | 85.55 | 88.77 | 66.43 | 49.00 | 62.39 |
| Qwen2.5-VL-72B | 1980 | 86.61 | **90.44** | 68.22 | 47.30 | 65.74 |
| Qwen-VL-Max | 2281 | 77.60 | 76.40 | – | – | 51.30 |
| Qwen2.5-Omni | **2340** | 81.80 | – | 64.0 | – | 64.30 |
| VITA | 2006.5 | 71.80 | – | 46.40 | – | 59.20 |
| LLaVA-OV-7B | – | 80.80 | – | 61.70 | – | 58.20 |
| LLaVA-OV-72B | – | 85.90 | – | 66.10 | 26.90 | 66.20 |
| InternVL-2-8B | – | 81.70 | – | 59.40 | – | – |
| InternVL-2-26B | – | 83.40 | – | 60.40 | – | – |
| Gemini-1.5-Pro | – | – | 70.90 | – | 33.10 | **75.00** |
| GPT-4V | 517/1409 | 75.00 | 74.30 | 57.10 | – | 59.90 |
| GPT-4o | 2310.3 | 83.10 | – | 64.70 | 34.70 | 71.90 |
| **Ours** | 1922 | **88.54** | 89.35 | **69.37** | **50.27** | 65.58 |

Table 2: Performance comparison on general multimodal and vision understanding tasks.

Our orchestration method achieves strong results on key benchmarks, with state-of-the-art performance on MMBench-EN (**88.54%**) and competitive performance on MMStar (**69.37%**) and LVBench (**50.27%**). Resource analysis shows intelligent task routing enables computational efficiency.

> **Finding 2:** Cross-modal memory pooling for context integration can enhance performance in complex visual tasks that require context awareness.

## 4.4 VIDEO UNDERSTANDING PERFORMANCE

Beyond static image tasks, we examine video understanding enhancement. Figure 3 shows significant gains across video lengths. For Qwen2-VL, we see +1.7%, +6.6%, and +12.9% improvements on short, medium, and long videos respectively (+7.0% overall). LLaVA-Video shows +3.3%, +6.2%, and +13.9% improvements (+7.8% overall), highlighting temporal reasoning capabilities.

## 4.5 TEXT-TO-SPEECH PROCESSING EFFICIENCY

Figure 4 shows our parallel processing reduces average time from 0.204s to 0.183s (10.3% reduction) while improving stability (standard deviation: 0.056s → 0.013s). These improvements demonstrate effective segmentation and parallel synthesis for real-time interaction.

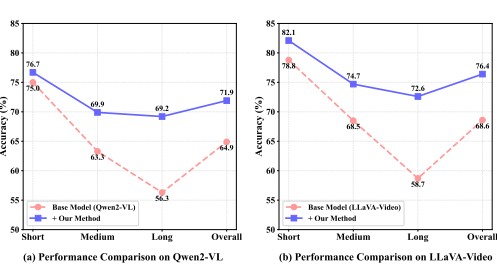

Figure 3: Performance comparison on Video-MME benchmark. The improvements are particularly significant for longer videos, where our controller's ability to maintain temporal context while integrating audio information proves most beneficial.

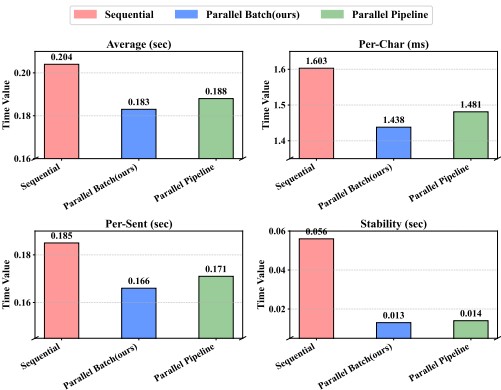

Figure 4: TTS processing architecture comparison showing significant improvements in both speed and stability with our parallel batch approaches.

## 5 CONCLUSIONS

In this paper, we present a novel, training-free orchestration framework for multimodal large language models, offering an innovative solution for the seamless integration of omni-modal fusion. Unlike traditional approaches that require extensive training data for feature-level fusion, our framework achieves elegant integration of multimodal capabilities through intelligent orchestration. The central controller LLM, with its excellent natural language understanding capabilities, automatically decomposes complex tasks and precisely invokes corresponding expert models through specific tokens. This flexible routing mechanism not only enables unified processing of multimodal inputs but also implements intelligent task decomposition and dynamic scheduling based on task characteristics. We innovatively designed a semantic-based batch processing TTS mechanism that significantly improves system response efficiency through intelligent segmentation, parallel processing, and result merging. Notably, our unified memory pool system, through standardized token encapsulation, successfully addresses the challenges faced by traditional systems in multimodal memory management. This design not only eliminates the inconvenience of switching between multiple independent models but also achieves coherent and natural multi-turn dialogue experiences through intelligent context management, providing users with an experience similar to interacting with a single super-LLM. A key advantage of our framework is that the expert model pool can be dynamically expanded or reduced at any time without requiring system retraining, offering exceptional flexibility in adapting to new capabilities and requirements. We believe this LLM orchestration-based approach will open up new possibilities for building more intelligent and natural human-computer interaction systems.

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
