APPENDIX

A. MLLM ORCHESTRATION ALGORITHM

This section presents the complete algorithm of our MLLM Orchestration framework, which details the step-by-step process of multimodal interaction handling, from speech input processing to final response generation.

---

**Algorithm 1** MLLM Orchestration Algorithm

---

**Require:** User speech input $s_t$, Cross-modal memory $M_{t-1}$
**Ensure:** Speech response $a_t$
 1: **procedure** ORCHESTRATIONPIPELINE($s_t, M_{t-1}$)
 2:     $q_t \leftarrow f_{\text{ASR}}(s_t)$                                   ▷ Speech recognition
 3:     $Y_t \leftarrow f_{\text{ctrl}}(q_t, M_{t-1})$                         ▷ Controller LLM reasoning
 4:     $C_t, S_t \leftarrow \text{SplitTokens}(Y_t)$                 ▷ Split content and control tokens
 5:     **if** $[S.stop] \in S_t$ **then**
 6:         CLEARTTSQUEUE
 7:         $M_t \leftarrow M_{t-1} \cup \{q_t, \text{interrupt\_flag}\}$
 8:         **return**
 9:     **if** $[S.listen] \in S_t$ **then**
10:         $M_t \leftarrow M_{t-1} \cup \{q_t, \text{await\_completion}\}$
11:         **return**
12:     $\mathcal{M} \leftarrow \delta_{\text{modality}}(S_t)$                   ▷ Modality selection
13:     $D_t \leftarrow \emptyset$                                       ▷ Initialize multimodal data collection
14:     **for** $m_i \in \mathcal{M}$ **do**
15:         **if** $m_i = $ vision **then**
16:             $v_t \leftarrow h_{\text{vision}}(q_t, M_{t-1})$
17:             $D_t \leftarrow D_t \cup \{v_t\}$
18:         **else if** $m_i = $ reasoning **then**
19:             $r_t \leftarrow h_{\text{reasoning}}(q_t, M_{t-1})$
20:             $D_t \leftarrow D_t \cup \{r_t\}$
21:     $\widetilde{Y}_t \leftarrow I(Y_t, D_t)$                                 ▷ Multimodal response fusion
22:     $T \leftarrow \text{segment}(\widetilde{Y}_t)$                     ▷ Semantic segmentation, see Appendix E
23:     **for** $i \leftarrow 1$ **to** $|T|$ **parallel do**
24:         $a_{t,i} \leftarrow g(T_i)$                                     ▷ Parallel TTS synthesis
25:     $a_t \leftarrow \text{stream\_concat}(a_{t,1}, \dots, a_{t,|T|})$         ▷ Stream concatenation
26:     $M_t \leftarrow h_{\text{compress}}(M_{t-1} \cup \{q_t, D_t, \widetilde{Y}_t\})$     ▷ Memory compression
27:     **return** $a_t$
28: **function** $\delta_{\text{MODALITY}}(S_t)$
29:     **return** $\{m_i | [S.need\_m_i] \in S_t, m_i \in \mathcal{M}\}$       ▷ $\mathcal{M}$ is set of available modalities
30: **function** SPLITTOKENS($Y_t$)
31:     $C_t \leftarrow \{y_i | y_i \in Y_t, y_i \text{ is content token}\}$
32:     $S_t \leftarrow \{y_i | y_i \in Y_t, y_i \text{ is control token}\}$
33:     **return** $C_t, S_t$

---

B. CORE INSTRUCTION SET

This section details the core instruction set of the MLLM Orchestration framework, serving as the fundamental communication protocol between the controller LLM and specialized models. The instruction set is designed with three primary objectives:

- **Semantic Clarity**: Each instruction has unambiguous meaning and predictable behavior
- **Dynamic Coordination**: Enables real-time task routing across heterogeneous modalities
- **Architectural Extensibility**: Supports seamless integration of new modalities through standardized patterns

Table 1: Core Instruction Set for Multimodal Interaction Control

| Instruction | Functionality and Usage Context |
| --- | --- |
| `[S.stop]` | Immediate process termination and resource release |
| `[S.need_vision]` | Activates visual processing pipeline (object detection, scene parsing) |
| `[S.need_reasoning]` | Invokes logical inference and causal analysis modules |
| `[S.speak]` | Generates verbal response with prosody control |
| `[S.listen]` | Enters passive listening mode for continuous input |
| `[S.need_*]` | Extensible pattern for future modality integration |

C. CORE PROMPT DESIGN

The prompt architecture enables natural human-machine interaction through three key design principles:

- **Role Specialization**: Clear separation of system responsibilities
- **Response Optimization**: Balanced information density and conversational flow
- **Modality Routing**: Intelligent resource allocation based on input analysis

```
Controller LLM prompt:  You are an AI assistant responsible for
coordinating specialized models.  Your primary functions include:

(1) Analyzing user intent through multimodal inputs.

      Output special control token [S.listen] if user input is
      incomplete or requires further clarification.
      Output special control token [S.stop] if the user intends to
      interrupt.
      Output special control token [S.speak] when a response is
      needed from you.

(2) Dynamically routing tasks to appropriate expert models.

      Use [S.need_vision] for visual queries.
      Use [S.need_reasoning] for analytical tasks.

(3) All responses must adhere to the format "Special Control Token +
    Response Content"; other response formats will be rejected.
```

D. MEMORY POOL SCHEMA

The cross-modal memory system employs a JSON-based schema with the following characteristics:

- **Structural Integrity**: Unified data representation across modalities

- **Retrieval Efficiency**: Optimized indexing for low-latency access
- **Context Preservation**: Long-term conversation state maintenance

**Key Architectural Features**:

- Modular design supporting pluggable extensions
- Content-aware compression algorithms
- Temporal and semantic indexing layers
- Version-controlled memory revisions

```
Memory JSON:
 {
     "id": "uuid4",
     "modality": "vision|audio|text|reasoning",
     "content": {
         "type": "mime_type",
         "data": "encrypted_blob",
         "embedding": "float_vector",
         "metadata": {
             "source": "input_device",
             "timestamp": "iso8601",
             "context": "semantic_tags"
         }
     },
     "turn_id": "sequential_counter",
     "references": ["related_uuids"],
     "priority": "0.0-1.0",
     "compression": {
         "algorithm": "zstd|lz4",
         "ratio": "float"
     }
 }
```

E. TTS SEGMENTATION RULES

The TTS module employs a rule-based segmentation strategy to optimize speech synthesis quality and latency. The segmentation rules are designed to maintain semantic coherence while enabling parallel processing. Below are the detailed segmentation rules:

```
Rule 1:  Natural punctuation boundaries | e.g., periods, commas,
semicolons
Rule 2:  Discourse markers and conjunctions | e.g., "however",
"therefore", "and"
Rule 3:  Syntactic boundaries | e.g., subject-predicate splits,
subordinate clauses
```

Each segment typically contains 7-15 words to balance synthesis quality and parallelization efficiency. This granularity ensures both semantic completeness and natural prosody while maintaining computational efficiency. For example, the sentence "The cat sat on the mat, while the dog slept peacefully" would be split into two segments: "The cat sat on the mat" and "while the dog slept peacefully".

F. STATEMENT ON LARGE LANGUAGE MODEL USAGE

In accordance with the ICLR 2026 policy on the use of Large Language Models (LLMs), we disclose that an LLM was utilized as a general-purpose writing assistance tool in the preparation of

this manuscript. The use of the LLM was strictly limited to post-writing text refinement with the following specific applications:

- **Grammar and Spelling Correction**: Identifying and correcting grammatical errors and typographical mistakes throughout the manuscript
- **Clarity and Diction Enhancement**: Suggesting alternative phrasing, refining word choices, and improving the overall readability of sentences and paragraphs that were already written by the authors
- **LaTeX Formatting Assistance**: Providing support with minor LaTeX code adjustments and debugging for proper formatting and presentation

**Scope Limitations and Author Responsibility**:

Crucially, the LLM was *not* used for research ideation, generating scientific claims, producing experimental results, or drafting any substantive parts of the paper from scratch. All concepts, methodologies, analyses, and conclusions presented in this work are entirely those of the human authors. The authors have carefully reviewed and edited all LLM-suggested modifications and take full responsibility for the scientific integrity and accuracy of the final submitted content.

This disclosure ensures full transparency regarding the limited and appropriate use of AI assistance tools in academic writing, maintaining the highest standards of research integrity and authorship accountability.

## G. ETHICS STATEMENT

This work adheres to the ICLR Code of Ethics and maintains the highest standards of research integrity throughout the investigation. The following ethical considerations have been carefully addressed:

- **Human and Animal Subjects**: No human subjects or animal experimentation was involved in this study
- **Data Privacy and Compliance**: All datasets used were sourced in compliance with relevant usage guidelines, ensuring no violation of privacy policies or data protection regulations
- **Bias Prevention**: We have taken systematic care to avoid any biases or discriminatory outcomes throughout our research process, including dataset selection, model evaluation, and result interpretation
- **Information Security**: No personally identifiable information was used, and no experiments were conducted that could raise privacy or security concerns

**Research Transparency and Integrity**:

We are committed to maintaining complete transparency and integrity throughout the research process. All experimental procedures, data handling protocols, and analytical methods have been designed to uphold ethical research standards. The multimodal orchestration framework proposed in this work is designed for beneficial applications in human-computer interaction, with careful consideration of potential societal impacts.

The authors declare no conflicts of interest that could inappropriately influence this research. All contributions and acknowledgments have been properly attributed, and the research has been conducted in accordance with institutional and international ethical guidelines for AI research.

## H. REPRODUCIBILITY STATEMENT

We have made comprehensive efforts to ensure that the results presented in this paper are fully reproducible by the research community. The following measures have been implemented to support replication and verification:

- **Code and Implementation**: All source code for the MLLM orchestration framework, including the LangGraph pipeline, modality-specific modules, and evaluation scripts, has been made publicly available in an anonymous repository to facilitate replication

- **Experimental Configuration**: Detailed descriptions of the experimental setup are provided, including training procedures, model configurations, hyperparameter settings, and hardware specifications used in our experiments
- **Dataset Accessibility**: All evaluation datasets used in this study, including MMBench, Video-MME, MMMU, and other benchmark suites, are publicly available and consistently accessible to ensure reproducible evaluation results
- **Model Specifications**: Complete specifications of all expert models employed in the orchestration framework are documented, including model versions, API endpoints, and configuration parameters

**Implementation Details and Resources**:

The orchestration framework implementation includes comprehensive documentation covering:

- Algorithm pseudocode and control flow specifications (Appendix A)
- Core instruction set and communication protocols (Appendix B)
- Prompt templates and dialogue control frameworks (Appendix C)
- Memory pool schema and cross-modal data management (Appendix D)
- TTS segmentation rules and parallel processing configurations (Appendix E)

We believe these comprehensive measures will enable other researchers to reproduce our work, validate our findings, and further advance research in training-free multimodal orchestration systems. All experimental conditions, evaluation metrics, and analysis procedures have been designed with reproducibility as a core principle.